# Learning To Explore Using Active Neural SLAM

**Devendra Singh Chaplot**[1][†]**, Dhiraj Gandhi**[2]**, Saurabh Gupta**[3][*]**,**
**Abhinav Gupta** [1,2][*]**, Ruslan Salakhutdinov**[1][*]
[1]Carnegie Mellon University, [2]Facebook AI Research, [3]UIUC

Project webpage: `https://devendrachaplot.github.io/projects/Neural-SLAM`
Code: `https://github.com/devendrachaplot/Neural-SLAM`

## ABSTRACT

This work presents a modular and hierarchical approach to learn policies for exploring 3D environments, called 'Active Neural SLAM'. Our approach leverages the strengths of both classical and learning-based methods, by using analytical path planners with learned SLAM module, and global and local policies. The use of learning provides flexibility with respect to input modalities (in the SLAM module), leverages structural regularities of the world (in global policies), and provides robustness to errors in state estimation (in local policies). Such use of learning within each module retains its benefits, while at the same time, hierarchical decomposition and modular training allow us to sidestep the high sample complexities associated with training end-to-end policies. Our experiments in *visually* and *physically* realistic simulated 3D environments demonstrate the effectiveness of our approach over past learning and geometry-based approaches. The proposed model can also be easily transferred to the PointGoal task and was the winning entry of the CVPR 2019 Habitat PointGoal Navigation Challenge.

## 1 INTRODUCTION

Navigation is a critical task in building intelligent agents. Navigation tasks can be expressed in many forms, for example, point goal tasks involve navigating to specific coordinates and semantic navigation involves finding the path to a specific scene or object. Irrespective of the task, a core problem for navigation in unknown environments is exploration, *i.e.*, how to efficiently visit as much of the environment. This is useful for maximizing the coverage to give the best chance of finding the target in unknown environments or for efficiently pre-mapping environments on a limited time-budget.

Recent work from Chen et al. (2019) has used end-to-end learning to tackle this problem. Their motivation is three-fold: *a)* learning provides flexibility to the choice of input modalities (classical systems rely on observing geometry through the use of specialized sensors, while learning systems can infer geometry directly from RGB images), *b)* use of learning can improve robustness to errors in explicit state estimation, and *c)* learning can effectively leverage structural regularities of the real world, leading to more efficient behavior in previously unseen environments. This lead to their design of an end-to-end trained neural network-based policy that processed raw sensory observations to directly output actions that the agent should execute.

While the use of learning for exploration is well-motivated, casting the exploration problem as an end-to-end learning problem has its own drawbacks. Learning about mapping, state-estimation, and path-planning purely from data in an end-to-end manner can be prohibitively expensive. Consequently, past end-to-end learning work for exploration from Chen et al. (2019) relies on the use of imitation learning and many millions of frames of experience, but still performs worse than classical methods that don't require any training at all.

This motivates our work. In this paper, we investigate alternate formulations of employing learning for exploration that retains the advantages that learning has to offer, but doesn't suffer from the

---

[†]Correspondence: `chaplot@cs.cmu.edu`
[*]Equal Contribution

drawbacks of full-blown end-to-end learning. Our key conceptual insight is that use of learning for leveraging structural regularities of indoor environments, robustness to state-estimation errors, and flexibility with respect to input modalities, happens at different time scales and can thus be factored out. This motivates the use of learning in a modular and hierarchical fashion inside of what one may call a 'classical navigation pipeline'. This results in navigation policies that can work with raw sensory inputs such as RGB images, are robust to state estimation errors, and leverage the regularities of real-world layouts. This results in extremely competitive performance over both geometry-based methods and recent learning-based methods; at the same time requiring a fraction of the number of samples.

More specifically, our proposed exploration architecture comprises of a learned Neural SLAM module, a global policy, and a local policy, that are interfaced via the map and an analytical path planner. The learned Neural SLAM module produces free space maps and estimates agent pose from input RGB images and motion sensors. The global policy consumes this free-space map with the agent pose and employs learning to exploit structural regularities in layouts of real-world environments to produce long-term goals. These long-term goals are used to generate short-term goals for the local policy (using a geometric path-planner). This local policy uses learning to directly map raw RGB images to actions that the agent should execute. Use of learning in the SLAM module provides flexibility with respect to input modality, learned global policy can exploit regularities in layouts of real-world environments, while learned local policies can use visual feedback to exhibit more robust behavior. At the same time, hierarchical and modular design and use of analytical planning, significantly cuts down the search space during training, leading to better performance as well as sample efficiency.

We demonstrate our proposed approach in *visually* and *physically* realistic simulators for the task of geometric exploration (visit as much area as possible). We work with the Habitat simulator from Savva et al. (2019). While Habitat is already visually realistic (it uses real-world scans from Chang et al. (2017) and Xia et al. (2018) as environments), we improve its physical realism by using actuation and odometry sensor noise models, that we collected by conducting physical experiments on a real mobile robot. Our experiments and ablations in this realistic simulation reveal the effectiveness of our proposed approach for the task of exploration. A straightforward modification of our method also tackles point-goal navigation tasks, and won the AI Habitat challenge at CVPR2019 across all tracks.

## 2 RELATED WORK

Navigation has been well studied in classical robotics. There has been a renewed interest in the use of learning to arrive at navigation policies, for a variety of tasks. Our work builds upon concepts in classical robotics and learning for navigation. We survey related works below.

**Navigation Approaches.** Classical approaches to navigation break the problem into two parts: mapping and path planning. Mapping is done via simultaneous localization and mapping (Thrun et al., 2005; Hartley and Zisserman, 2003; Fuentes-Pacheco et al., 2015), by fusing information from multiple views of the environment. While sparse reconstruction can be done well with monocular RGB images (Mur-Artal and Tardós, 2017), dense mapping is inefficient (Newcombe et al., 2011) or requires specialized scanners such as Kinect (Izadi et al., 2011). Maps are used to compute paths to goal locations via path planning (Kavraki et al., 1996; Lavalle and Kuffner Jr, 2000; Canny, 1988). These classical methods have inspired recent learning-based techniques. Researchers have designed neural network policies that reason via spatial representations (Gupta et al., 2017; Parisotto and Salakhutdinov, 2018; Zhang et al., 2017; Henriques and Vedaldi, 2018; Gordon et al., 2018), topological representations (Savinov et al., 2018; 2019), or use differentiable and trainable planners (Tamar et al., 2016; Lee et al., 2018; Gupta et al., 2017; Khan et al., 2017). Our work furthers this research, and we study a hierarchical and modular decomposition of the problem and employ learning inside these components instead of end-to-end learning. Research also focuses on incorporating semantics in SLAM (Pronobis and Jensfelt, 2012; Walter et al., 2013).

**Exploration in Navigation.** While a number of works focus on passive map-building, path planning and goal-driven policy learning, a much smaller body of work tackles the the problem of active SLAM, i.e., how to actively control the camera for map building. We point readers to Fuentes-Pacheco et al. (2015) for a detailed survey, and summarize the major themes below. Most such works frame this problem as a Partially Observable Markov Decision Process (POMDP) that are approximately solved (Martinez-Cantin et al., 2009; Kollar and Roy, 2008), and or seek to find a sequence of actions that minimizes uncertainty of maps (Stachniss et al., 2005; Carlone et al., 2014).

Another line of work explores by picking vantage points (such as on the frontier between explored and unexplored regions (Dornhege and Kleiner, 2013; Holz et al., 2010; Yamauchi, 1997; Xu et al., 2017)). Recent works from Chen et al. (2019); Savinov et al. (2019); Fang et al. (2019) attack this problem via learning. Our proposed modular policies unify the last two lines of research, and we show improvements over representative methods from both these lines of work. Exploration has also been studied more generally in RL in the context of exploration-exploitation trade-off (Sutton and Barto, 2018; Kearns and Singh, 2002; Auer, 2002; Jaksch et al., 2010).

**Hierarchical and Modular Policies.** Hierarchical RL (Dayan and Hinton, 1993; Sutton et al., 1999; Barto and Mahadevan, 2003) is an active area of research, aimed at automatically discovering hierarchies to speed up learning. However, this has proven to be challenging, and thus most work has resorted to using hand-defining hierarchies. For example in the context of navigation, Bansal et al. (2019) and Kaufmann et al. (2019) design modular policies for navigation, that interface learned policies with low-level feedback controllers. Hierarchical and modular policies have also been used for Embodied Question Answering (Das et al., 2018a; Gordon et al., 2018; Das et al., 2018b).

## 3 TASK SETUP

We follow the exploration task setup proposed by Chen et al. (2019) where the objective is to maximize the coverage in a fixed time budget. The coverage is defined as the total area in the map known to be traversable. Our objective is to train a policy which takes in an observation $s_t$ at each time step $t$ and outputs a navigational action $a_t$ to maximize the coverage.

We try to make our experimental setup in simulation as realistic as possible with the goal of transferring trained policies to the real world. We use the Habitat simulator (Savva et al., 2019) with the Gibson (Xia et al., 2018) and Matterport (MP3D) (Chang et al., 2017) datasets for our experiments. Both Gibson and Matterport datasets are based on real-world scene reconstructions are thus significantly more realistic than synthetic SUNCG dataset (Song et al., 2017) used for past research on exploration (Chen et al., 2019; Fang et al., 2019).

In addition to synthetic scenes, prior works on learning-based navigation have also assumed simplistic agent motion. Some works limit agent motion on a grid with 90 degree rotations (Zhu et al., 2017; Gupta et al., 2017; Chaplot et al., 2018). Other works which implement fine-grained control, typically assume unrealistic agent motion with no noise (Savva et al., 2019) or perfect knowledge of agent pose (Chaplot et al., 2016). Since the motion is simplistic, it becomes trivial to estimate the agent pose in most cases even if it is not assumed to be known. The reason behind these assumptions on agent motion and pose is that motion and sensor noise models are not known. In order to relax both these assumptions, we collect motion and sensor data in the real-world and implement more realistic agent motion and sensor noise models in the simulator as described in the following subsection.

### 3.1 ACTUATION AND SENSOR NOISE MODEL

We represent the agent pose by $(x, y, o)$ where $x$ and $y$ represent the $xy$ co-ordinate of the agent measured in metres and $o$ represents the orientation of the agent in radians (measured counterclockwise from $x$-axis). Without loss of generality, assume agents starts at $p_0 = (0, 0, 0)$. Now, suppose the agent takes an action $a_t$. Each action is implemented as a control command on a robot. Let the corresponding control command be $\Delta u_a = (x_a, y_a, o_a)$. Let the agent pose after the action be $p_1 = (x^\star, y^\star, o^\star)$. The actuation noise ($\epsilon_{act}$) is the difference between the actual agent pose ($p_1$) after the action and the intended agent pose ($p_0 + \Delta u$):

$$\epsilon_{act} = p_1 - (p_0 + \Delta u) = (x^\star - x_a, y^\star - y_a, o^\star - o_a)$$

Mobile robots typically have sensors which estimate the robot pose as it moves. Let the sensor estimate of the agent pose after the action be $p_1' = (x', y', o')$. The sensor noise ($\epsilon_{sen}$) is given by the difference between the sensor pose estimate ($p_1'$) and the actual agent pose($p_1$):

$$\epsilon_{sen} = p_1' - p_1 = (x' - x^\star, y' - y^\star, o' - o^\star)$$

In order to implement the actuation and sensor noise models, we would like to collect data for navigational actions in the Habitat simulator. We use three default navigational actions: Forward: move forward by 25cm, Turn Right: on the spot rotation clockwise by 10 degrees, and Turn Left: on the spot rotation counter-clockwise by 10 degrees. The control commands are implemented as

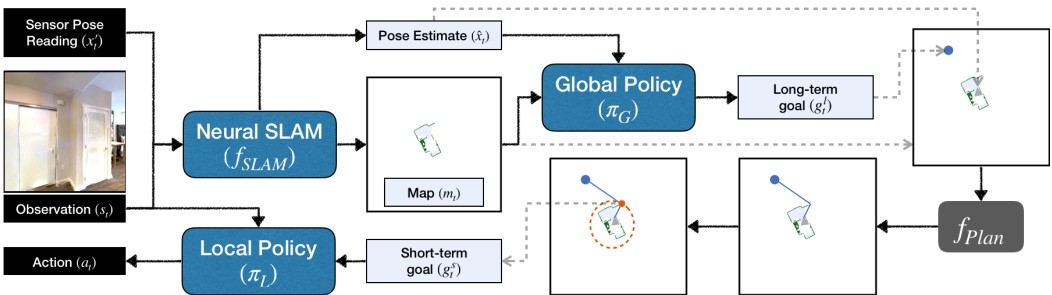

**Figure 1: Overview of our approach**. The Neural SLAM module predicts a map and agent pose estimate from incoming RGB observations and sensor readings. This map and pose are used by a Global policy to output a long-term goal, which is converted to a short-term goal using an analytic path planner. A Local Policy is trained to navigate to this short-term goal.

$u_{Forward} = (0.25, 0, 0)$, $u_{Right} : (0, 0, -10 * \pi/180)$ and $u_{Left} : (0, 0, 10 * \pi/180)$. In practice, a robot can also rotate slightly while moving forward and translate a bit while rotating on-the-spot, creating rotational actuation noise in forward action and similarly, a translation actuation noise in on-the-spot rotation actions.

We use a LoCoBot[1] to collect data for building the actuation and sensor noise models. We use the pyrobot API (Murali et al., 2019) along with ROS (Quigley et al., 2009) to implement the control commands and get sensor readings. For each action $a$, we fit a separate Gaussian Mixture Model for the actuation noise and sensor noise, making a total of 6 models. Each component in these Gaussian mixture models is a multi-variate Gaussian in 3 variables, $x$, $y$ and $o$. For each model, we collect 600 datapoints. The number of components in each Gaussian mixture model is chosen using cross-validation. We implement these actuation and sensor noise models in the Habitat simulator for our experiments. We have released the noise models, along with their implementation in the Habitat simulator in the open-source code.

## 4 METHODS

We propose a modular navigation model, 'Active Neural SLAM'. It consists of three components: a *Neural SLAM module*, a *Global policy* and a *Local policy* as shown in Figure 1. The Neural SLAM module predicts the map of the environment and the agent pose based on the current observations and previous predictions. The Global policy uses the predicted map and agent pose to produce a long-term goal. The long-term goal is converted into a short-term goal using path planning. The Local policy takes navigational actions based on the current observation to reach the short-term goal.

**Map Representation.** The Active Neural SLAM model internally maintains a spatial map, $m_t$ and pose of the agent $x_t$. The spatial map, $m_t$, is a $2 \times M \times M$ matrix where $M \times M$ denotes the map size and each element in this spatial map corresponds to a cell of size $25cm^2$ ($5cm \times 5cm$) in the physical world. Each element in the first channel denotes the probability of an obstacle at the corresponding location and each element in the second channel denotes the probability of that location being explored. A cell is considered to be explored when it is known to be free space or an obstacle. The spatial map is initialized with all zeros at the beginning of an episode, $m_0 = [0]^{2 \times M \times M}$. The pose $x_t \in \mathbb{R}^3$ denotes the $x$ and $y$ coordinates of the agent and the orientation of the agent at time $t$. The agent always starts at the center of the map facing east at the beginning of the episode, $x_0 = (M/2, M/2, 0.0)$.

**Neural SLAM Module.** The Neural SLAM Module ($f_{SLAM}$) takes in the current RGB observation, $s_t$, the current and last sensor reading of the agent pose $x'_{t-1:t}$, last agent pose and map estimates, $\hat{x}_{t-1}, m_{t-1}$ and outputs an updated map, $m_t$, and the current agent pose estimate, $\hat{x}_t$, (see Figure 2): $m_t, \hat{x}_t = f_{SLAM}(s_t, x'_{t-1:t}, \hat{x}_{t-1}, m_{t-1}|\theta_S)$, where $\theta_S$ denote the trainable parameters of the Neural SLAM module. It consists of two learned components, a Mapper and a Pose Estimator. The Mapper ($f_{Map}$) outputs a egocentric top-down 2D spatial map, $p_t^{ego} \in [0, 1]^{2 \times V \times V}$ (where $V$ is the vision range), predicting the obstacles and the explored area in the current observation. The Pose Estimator ($f_{PE}$) predicts the agent pose ($\hat{x}_t$) based on past pose estimate ($\hat{x}_{t-1}$) and last two

---

[1]http://locobot.org

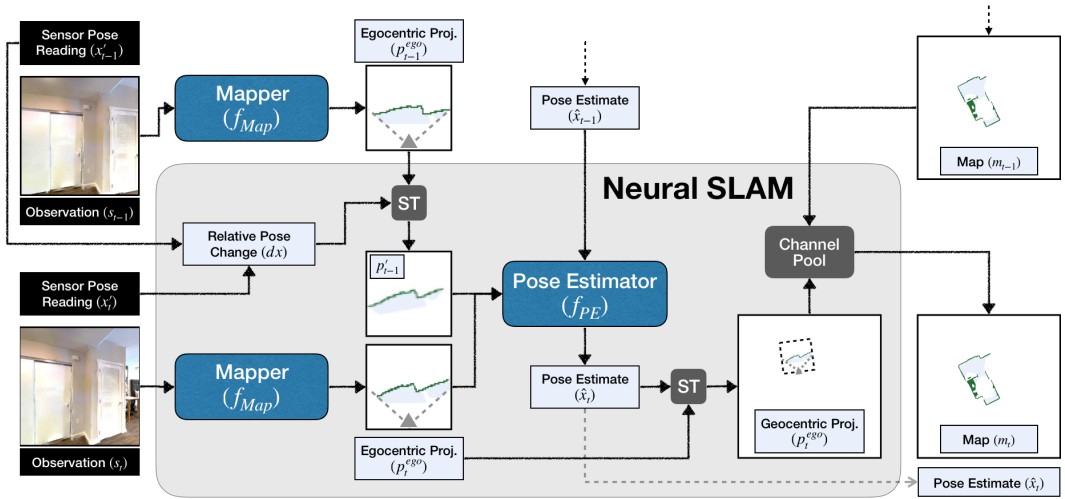

**Figure 2: Architecture of the Neural SLAM module:** The Neural SLAM module ($f_{Map}$) takes in the current RGB observation, $s_t$, the current and last sensor reading of the agent pose $x'_{t-1:t}$, last agent pose estimate, $\hat{x}_{t-1}$ and the map at the previous time step $m_{t-1}$ and outputs an updated map, $m_t$ and the current agent pose estimate, $\hat{x}_t$. 'ST' denotes spatial transformation.

egocentric map predictions ($p^{ego}_{t-1:t}$). It essentially compares the current egocentric map prediction to the last egocentric map prediction transformed to the current frame to predict the pose change between the two maps. The egocentric map from the Mapper is transformed to a geocentric map based on the pose estimate given by the Pose Estimator and then aggregated with the previous spatial map ($m_{t-1}$) to get the current map ($m_t$). More implementation details of the Neural SLAM module are provided in the Appendix.

**Global Policy.** The Global Policy takes $h_t \in [0, 1]^{4 \times M \times M}$ as input, where the first two channels of $h_t$ are the spatial map $m_t$ given by the SLAM module, the third channel represents the current agent position estimated by the SLAM module, the fourth channel represents the visited locations, i.e. $\forall i, j \in \{1, 2, \ldots, m\}$:

$$h_t[c, i, j] = m_t[c, i, j] \quad \forall c \in \{0, 1\}$$
$$h_t[2, i, j] = 1 \qquad \text{if } i = \hat{x}_t[0] \text{ and } j = \hat{x}_t[1]$$
$$h_t[3, i, j] = 1 \qquad \text{if } (i, j) \in [(\hat{x}_k[0], \hat{x}_k[1])]_{k \in \{0, 1, \ldots, t\}}$$

We perform two transformations before passing $h_t$ to the Global Policy model. The first transformation subsamples a window of size $4 \times G \times G$ around the agent from $h_t$. The second transformation performs max pooling operations to get an output of size $4 \times G \times G$ from $h_t$. Both the transformations are stacked to form a tensor of size $8 \times G \times G$ and passed as input to the Global Policy model. The Global Policy uses a convolutional neural network to predict a long-term goal, $g^l_t$ in $G \times G$ space: $g^l_t = \pi_G(h_t | \theta_G)$, where $\theta_G$ are the parameters of the Global Policy.

**Planner.** The Planner takes the long-term goal ($g^l_t$), the spatial obstacle map ($m_t$) and the agnet pose estimate ($\hat{x}_t$) as input and computes the short-term goal $g^s_t$, i.e. $g^s_t = f_{Plan}(g^l_t, m_t, \hat{x}_t)$. It computes the shortest path from the current agent location to the long-term goal ($g^l_t$) using the Fast Marching Method (Sethian, 1996) based on the current spatial map $m_t$. The unexplored area is considered as free space for planning. We compute a short-term goal coordinate (farthest point within $d_s (= 0.25m)$ from the agent) on the planned path.

**Local Policy.** The Local Policy takes as input the current RGB observation ($s_t$) and the short-term goal ($g^s_t$) and outputs a navigational action, $a_t = \pi_L(s_t, g^s_t | \theta_L)$, where $\theta_L$ are the parameters of the Local Policy. The short-term goal coordinate is transformed into relative distance and angle from the agent's location before being passed to the Local Policy. The Local Policy is a recurrent neural network consisting of a pretrained ResNet18 (He et al., 2016) as the visual encoder.

## 5 EXPERIMENTAL SETUP

We use the Habitat simulator (Savva et al., 2019) with the Gibson (Xia et al., 2018) and Matterport (MP3D) (Chang et al., 2017) datasets for our experiments. Both Gibson and MP3D consist of scenes

which are 3D reconstructions of real-world environments, however, Gibson is collected using a different set of cameras, consists mostly of office spaces while MP3D consists of mostly homes with a larger average scene area. We will use Gibson as our training domain, and use MP3D for domain generalization experiments. The observation space consists of RGB images of size $3 \times 128 \times 128$ and base odometry sensor readings of size $3 \times 1$ denoting the change in agent's x-y coordinates and orientation. The actions space consists of three actions: `move_forward`, `turn_left`, `turn_right`. Both the base odometry sensor readings and the agent motion based on the actions are noisy. They are implemented using the sensor and actuation noise models based on real-world data as discussed in Section 3.1.

We follow the Exploration task setup proposed by Chen et al. (2019) where the objective to maximize the coverage in a fixed time budget. Coverage is the total area in the map known to be traversable. We define a traversable point to be known if it is in the field-of-view of the agent and is less than $3.2m$ away. We use two evaluation metrics, the absolute coverage area in $m^2$ (**Cov**) and the percentage of area explored in the scene (**% Cov**), i.e. ratio of coverage to maximum possible coverage in the corresponding scene. During training, each episode lasts for a fixed length of 1000 steps.

We use train/val/test splits provided by Savva et al. (2019) for both the datasets. Note that the set of scenes used in each split is disjoint, which means the agent is tested on new scenes never seen during training. Gibson test set is not public but rather held out on an online evaluation server for the Pointgoal task. We use the validation as the test set for comparison and analysis for the Gibson domain. We do not use the validation set for hyper-parameter tuning. To analyze the performance of all the models with respect to the size of the scene, we split the Gibson validation set into two parts, a small set of 10 scenes with explorable area ranging from $16m^2$ to $36m^2$, and a large set of 4 scenes with explorable area ranging from $55m^2$ to $100m^2$. Note that the size of the map is usually much larger than the traversable area, with the largest map being about $23m$ long and $11m$ wide.

**Training Details.** We train our model in the Gibson domain and transfer it to the Matterport domain. The Mapper is trained to predict egocentric projections, and the Pose Estimator is trained to predict agent pose using supervised learning. The ground truth egocentric projection is computed using geometric projections from ground truth depth. The Global Policy is trained using Reinforcement Learning with reward proportional to the increase in coverage as the reward. The Local Policy is trained using Imitation Learning (behavioral cloning). All the modules are trained simultaneously. Their parameters are independent, but the data distribution is inter-dependent. Based on the actions taken by the Local policy, the future input to Neural SLAM module changes, which in turn changes the map and agent pose input to the Global policy and consequently affects the short-term goal given to the Local Policy. For more architecture and hyperparameter details, please refer to the supplementary material and the open-source code.

**Baselines.** We use a range of end-to-end Reinforcement Learning (RL) methods as baselines:
**RL + 3LConv:** An RL Policy with 3 layer convolutional network followed by a GRU (Cho et al., 2014) as described by Savva et al. (2019).
**RL + Res18:** A RL Policy initialized with ResNet18 (He et al., 2016) pre-trained on ImageNet followed by a GRU.
**RL + Res18 + AuxDepth:** This baseline is adapted from Mirowski et al. (2017) who use depth prediction as an auxiliary task. We use the same architecture as our Neural SLAM module (conv layers from ResNet18) with one additional deconvolutional layer for Depth prediction followed by 3 layer convolution and GRU for the policy.
**RL + Res18 + ProjDepth:** This baseline is adapted form Chen et al. (2019) who project the depth image in an egocentric top-down in addition to the RGB image as input to the RL policy. Since we do not have depth as input, we use the architecture from RL + Res18 + AuxDepth for depth prediction and project the predicted depth before passing to 3Layer Conv and GRU policy.

For all the baselines, we also feed a 32-dimensional embedding of the sensor pose reading to the GRU along with the image-based representation. This embedding is also learnt end-to-end using RL. All baselines are trained using PPO (Schulman et al., 2017) with increase in coverage as the reward (identical to the reward used for Global policy). All the baselines require access to the ground-truth map during training for computing the reward. The supervision for the Global Policy, the Local Policy and the Mapper can also be obtained from the ground-truth map. The Pose Estimator requires additional supervision in the form of the ground-truth agent pose. We study the effect of this additional supervision in ablation experiments.

,

**Table 1:** Exploration performance of the proposed model, Active Neural SLAM (ANS) and baselines. The baselines are adated from [1] Savva et al. (2019), [2] Mirowski et al. (2017) and [3] Chen et al. (2019).

| | Gibson Val | | Domain Generalization MP3D Test | |
|---|---|---|---|---|
| Method | % Cov. | Cov. (m2) | % Cov. | Cov. (m2) |
| RL + 3LConv [1] | 0.737 | 22.838 | 0.332 | 47.758 |
| RL + Res18 | 0.747 | 23.188 | 0.341 | 49.175 |
| RL + Res18 + AuxDepth [2] | 0.779 | 24.467 | 0.356 | 51.959 |
| RL + Res18 + ProjDepth [3] | 0.789 | 24.863 | 0.378 | 54.775 |
| **Active Neural SLAM (ANS)** | **0.948** | **32.701** | **0.521** | **73.281** |

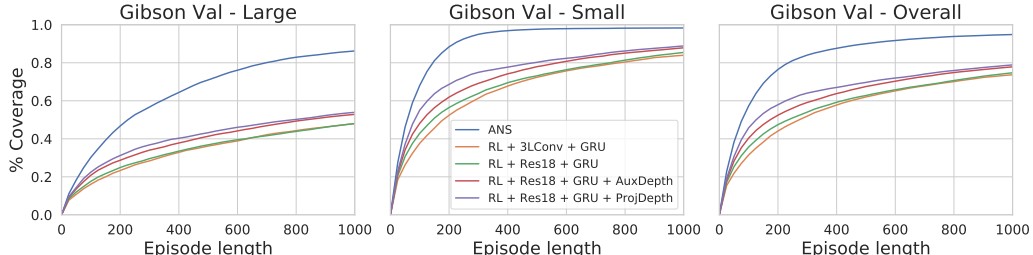

**Figure 3:** Plot showing the % Coverage as the episode progresses for ANS and the baselines on the large and small scenes in the Gibson Val set as well as the overall Gibson Val set.

## 6 RESULTS

We train the proposed ANS model and all the baselines for the Exploration task with 10 million frames on the Gibson training set. The results are shown in Table 1. The results on the Gibson Val set are averaged over a total of 994 episodes in 14 different unseen scenes. The proposed model achieves an average absolute and relative coverage of $32.701m^2/0.948$ as compared to $24.863m^2/0.789$ for the best baseline. This indicates that the proposed model is more efficient and effective at exhaustive exploration as compared to the baselines. This is because our hierarchical policy architecture reduces the horizon of the long-term exploration problem as instead of taking tens of low-level navigational actions, the Global policy only takes few long-term goal actions. We also report the domain generalization performance on the Exploration task in Table 1 (see shaded region), where all models trained on Gibson are evaluated on the Matterport domain. ANS leads to higher domain generalization performance ($73.281m^2/0.521$ vs $54.775m^2/0.378$). The absolute coverage is higher and % Cov is lower for the Matterport domain as it consists of larger scenes on average. On a set of small MP3D test scenes (comparable to Gibson scene sizes), ANS achieved a performance of $31.407m^2/0.836$ as compared to $23.091m^2/0.620$ for the best baseline. Some visualizations of policy execution are provided in Figure 4[2].

In Fig. 3, we plot the relative coverage (% Cov) of all the models as the episode progresses on the large and small scene sets, as well as the overall Gibson Val set. The plot on the small scene set shows that ANS is able to almost completely explore the small scenes in around 500 steps, however, the baselines are only able to explore 85-90% of the small scenes in 1000 steps (see Fig. 3 center). This indicates that ANS explores more efficiently in small scenes. The plot on the large scenes set shows that the performance gap between ANS and baselines widens as the episode progresses (see Fig. 3 left). Looking at the behavior of the baselines, we saw that they often got stuck in local areas. This behavior indicates that they are unable to remember explored areas over long-time horizons and are ineffective at long-term planning. On the other hand, ANS uses a Global policy on the map which allows it to have the memory of explored areas over long-time horizons, and plan effectively to reach distant long-term goals by leveraging analytical planners. As a result, it is able to explore effectively in large scenes with long episode lengths.

---

[2]See https://devendrachaplot.github.io/projects/Neural-SLAM for visualization videos.

**Table 2:** Results of the ablation experiments on the Gibson environment.

| Method | Gibson Val Overall | | Gibson Val Large | | Gibson Val Small | |
|---|---|---|---|---|---|---|
| | % Cov. | Cov. (m2) | % Cov. | Cov. (m2) | % Cov. | Cov. (m2) |
| ANS w/o Local Policy + Det. Planner | 0.941 | 32.188 | 0.845 | 53.999 | 0.980 | 23.464 |
| ANS w/o Global Policy + FBE | 0.925 | 30.981 | 0.782 | 49.731 | 0.982 | 23.481 |
| ANS w/o Pose Estimation | 0.916 | 30.746 | 0.771 | 49.518 | 0.973 | 23.237 |
| **ANS** | **0.948** | **32.701** | **0.862** | **55.608** | **0.983** | **23.538** |

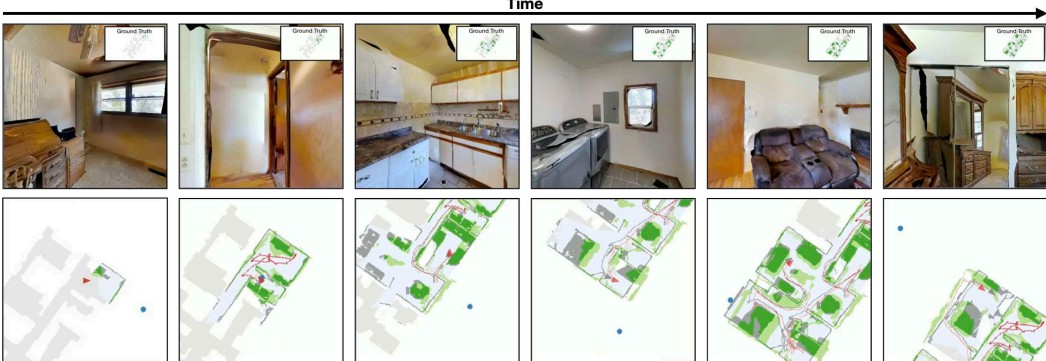

**Figure 4: Exploration visualization**. Figure showing a sample trajectory of the Active Neural SLAM model in the Exploration task. *Top:* RGB observations seen by the agent. *Inset:* Global ground truth map and pose (not visible to the agent). *Bottom:* Local map and pose predictions. Long-term goals selected by the Global policy are shown by blue circles. The ground-truth map and pose are under-laid in grey. Map prediction is overlaid in green, with dark green denoting correct predictions and light green denoting false positives. Agent pose predictions are shown in red. The light blue shaded region shows the explored area.

## 6.1 ABLATIONS

**Local Policy.** An alternative to learning a Local Policy is to have a deterministic policy which follows the plan given by the Planner. As shown in Table 2, the ANS model performs slightly worse without the Local Policy. The Local Policy is designed to adapt to small errors in Mapping. We observed Local policy overcoming false positives encountered in mapping. For example, the Neural SLAM module could sometime wrongly predict a carpet as an obstacle. In this case, the planner would plan to go around the carpet. However, if the short-term goal is beyond the carpet, the Local policy can understand that the carpet is not an obstacle based on the RGB observation and learn to walk over it.

**Global Policy.** An alternative to learning a Global Policy for sampling long-term goals is to use a classical algorithm called Frontier-based exploration (FBE) (Yamauchi, 1997). A frontier is defined as the boundary between the explored free space and the unexplored space. Frontier-based exploration essentially sample points on this frontier as goals to explore the space. There are different variants of Frontier-based exploration based on the sampling strategy. Holz et al. (2010) compare different sampling strategies and find that sampling the point on the frontier closest to the agent gives the best results empirically. We implement this variant and replace it with our learned Global Policy. As shown in Table 2, the performance of the Frontier-based exploration policy is comparable on small scenes, but around 10% lower on large scenes, relative to the Global policy. This indicates the importance of learning as compared to classical exploration methods in larger scenes. Qualitatively, we observed that Frontier-based exploration spent a lot of time exploring corners or small areas behind furniture. In contrast, the trained Global policy ignored small spaces and chose distant long-term goals which led to higher coverage.

**Pose Estimation.** A difference between ANS and the baselines is that ANS uses additional supervision to train the Pose Estimator. In order to understand whether the performance gain is coming from this additional supervision, we remove the Pose Estimator from ANS and just use the input sensor reading as our pose estimate. Results in Table 2 show that the ANS still outperforms the baselines even without the Pose Estimator. We also observed that performance without the pose estimator drops only about 1% on small scenes, but around 10% on large scenes. This is expected because larger scenes take longer to explore, and pose errors accumulate over time to cause drift. Passing the ground truth pose as input the baselines instead of the sensor reading did not improve their performance.

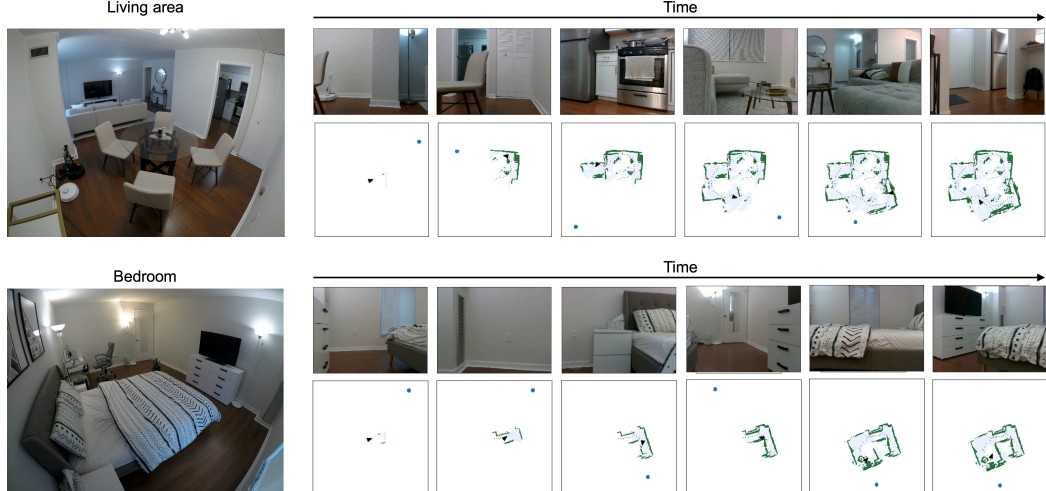

**Figure 5: Real-world Transfer. Left:** Image showing the living area in an apartment used for the real-world experiments. **Right:** Sample images seen by the robot and the predicted map. The long-term goal selected by the Global Policy is shown by a blue circle on the map.

## 6.2 REAL-WORLD TRANSFER

We deploy the trained ANS policy on a Locobot in the real-world. In order to match the real-world observations to the simulator observations as closely as possible, we change the simulator input configuration to match the camera intrinsics on the Locobot. This includes the camera height and horizontal and vertical field-of-views. In Figure 5, we show an episode of ANS exploring the living area in an apartment. The figure shows that the policy transfers well to the real-world and is able to effectively explore the environment. The long-term goals sampled by the Global policy (shown by blue circles on the map) are often towards open spaces in the explored map, which indicates that it is learning to exploit the structure in the map. Please refer to the project webpage for real-world transfer videos.

## 6.3 POINTGOAL TASK TRANSFER.

PointGoal has been the most studied task in recent literature on navigation where the objective is to navigate to a goal location whose relative coordinates are given as input in a limited time budget. In this task, each episode ends when either the agent takes the `stop` action or at a maximum of 500 timesteps. An episode is considered a success when the final position of the agent is within 0.2m of the goal location. In addition to Success rate (Succ), Success weighted by (normalized inverse) Path Length or SPL is also used as a metric for evaluation as proposed by Anderson et al. (2018).

All the baseline models trained for the task of Exploration either need to be retrained or at least fine-tuned to be transferred to the Pointgoal task. The modularity of ANS provides it another advantage that it can be transferred to the Pointgoal task without any additional training. For transferring to the Pointgoal task, we just fix the Global policy to always output the PointGoal coordinates as the long-term goal and use the Local Policy and Neural SLAM module trained for the Exploration task. We found that an ANS policy trained on exploration, when transferred to the Pointgoal task performed better than several RL and Imitation Learning baselines trained on the Pointgoal task. The transferred ANS model achieves a success rate/SPL of 0.950/0.846 as compared to 0.827/0.730 for the best baseline model on Gibson val set. The ANS model also generalized significantly better than the baselines to harder goals and to the Matterport domain. In addition to better performance, ANS was also 10 to 75 times more sample efficient than the baselines. This transferred ANS policy was also the winner of the CVPR 2019 Habitat Pointgoal Navigation Challenge for both RGB and RGB-D tracks among over 150 submissions from 16 teams. These results highlight a key advantage of our model. It allows us to transfer the knowledge of obstacle avoidance and control in low-level navigation across tasks, as the Local Policy and Neural SLAM module are task-invariant. More details about the Pointgoal experiments, baselines, results including domain and goal generalization on the Pointgoal task are provided in the supplementary material.

## 7 CONCLUSION

In this paper, we proposed a modular navigational model which leverages the strengths of classical and learning-based navigational methods. We show that the proposed model outperforms prior methods on both Exploration and PointGoal tasks and shows strong generalization across domains, goals, and tasks. In the future, the proposed model can be extended to complex semantic tasks such as Semantic Goal Navigation and Embodied Question Answering by using a semantic Neural SLAM module which creates a multi-channel map capturing semantic properties of the objects in the environment. The model can also be combined with prior work on Localization to relocalize in a previously created map for efficient navigation in subsequent episodes.

## ACKNOWLEDGEMENTS

This work was supported by IARPA DIVA D17PC00340, ONR Grant N000141812861, ONR MURI, ONR Young Investigator, DARPA MCS, and Apple. We would also like to acknowledge NVIDIA's GPU support. We thank Guillaume Lample for discussions and coding during the initial stages of this project.

**Licenses for referenced datasets.**
Gibson: http://svl.stanford.edu/gibson2/assets/GDS_agreement.pdf
Matterport3D: http://kaldir.vc.in.tum.de/matterport/MP_TOS.pdf

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

**Table 3:** Performance of the proposed model, Active Neural SLAM (ANS) and all the baselines on the Exploration task. 'ANS - Task Transfer' refers to the ANS model transferred to the PointGoal task after training on the Exploration task.

| | | | | Domain Generalization | | Goal Generalization | | | |
| | | | | MP3D Test | | Hard-GEDR | | Hard-Dist | |
| Test Setting → | | Gibson Val | | | | | | | |
| Train Task | Method | Succ | SPL | Succ | SPL | Succ | SPL | Succ | SPL |
|---|---|---|---|---|---|---|---|---|---|
| PointGoal | Random | 0.027 | 0.021 | 0.010 | 0.010 | 0.000 | 0.000 | 0.000 | 0.000 |
| | RL + Blind | 0.625 | 0.421 | 0.136 | 0.087 | 0.052 | 0.020 | 0.008 | 0.006 |
| | RL + 3LConv + GRU | 0.550 | 0.406 | 0.102 | 0.080 | 0.072 | 0.046 | 0.006 | 0.006 |
| | RL + Res18 + GRU | 0.561 | 0.422 | 0.160 | 0.125 | 0.176 | 0.109 | 0.004 | 0.003 |
| | RL + Res18 + GRU + AuxDepth | 0.640 | 0.461 | 0.189 | 0.143 | 0.277 | 0.197 | 0.013 | 0.011 |
| | RL + Res18 + GRU + ProjDepth | 0.614 | 0.436 | 0.134 | 0.111 | 0.180 | 0.129 | 0.008 | 0.004 |
| | IL + Res18 + GRU | 0.823 | 0.725 | 0.365 | 0.318 | 0.682 | 0.558 | 0.359 | 0.310 |
| | CMP | 0.827 | 0.730 | 0.320 | 0.270 | 0.670 | 0.553 | 0.369 | 0.318 |
| | ANS | **0.951** | **0.848** | **0.593** | **0.496** | **0.824** | **0.710** | 0.662 | **0.534** |
| Exploration | ANS - Task Transfer | 0.950 | 0.846 | 0.588 | 0.490 | 0.821 | 0.703 | **0.665** | 0.532 |

## A  POINTGOAL EXPERIMENTS

PointGoal has been the most studied task in recent literature on navigation where the objective is to navigate to a goal location whose relative coordinates are given as input in a limited time budget. We follow the PointGoal task setup from Savva et al. (2019), using train/val/test splits for both Gibson and Matterport datasets. Note that the set of scenes used in each split is disjoint, which means the agent is tested on new scenes never seen during training. Gibson test set is not public but rather held out on an online evaluation server[3]. We report the performance of our model on the Gibson test set when submitted to the online server but also use the validation set as another test set for extensive comparison and analysis. We do not use the validation set for hyper-parameter tuning.

Savva et al. (2019) identify two measures to quantify the difficulty of a PointGoal dataset. The first is the average geodesic distance (distance along the shortest path) to the goal location from the starting location of the agent, and the second is the average geodesic to Euclidean distance ratio (GED ratio). The GED ratio is always greater than or equal to 1, with higher ratios resulting in harder episodes. The train/val/test splits in the Gibson dataset come from the same distribution of having similar average geodesic distance and GED ratio. In order to analyze the performance of the proposed model on out-of-set goal distribution, we create two harder sets, Hard-Dist and Hard-GEDR. In the Hard-Dist set, the geodesic distance to goal is always more than 10m and the average geodesic distance to the goal is 13.48m as compared to $6.9/6.5/7.0$m in train/val/test splits (Savva et al., 2019). Hard-GEDR set consists of episodes with an average GED ratio of 2.52 and a minimum GED ratio of 2.0 as compared to average GED ratio 1.37 in the Gibson val set.

We also follow the episode specification from Savva et al. (2019). Each episode ends when either the agent takes the `stop` action or at a maximum of 500 timesteps. An episode is considered a success when the final position of the agent is within 0.2m of the goal location. In addition to Success rate (Succ), we also use Success weighted by (normalized inverse) Path Length or SPL as a metric for evaluation for the PointGoal task as proposed by Anderson et al. (2018).

### A.1  POINTGOAL RESULTS

In Table 3, we show the performance of the proposed model transferred to the PointGoal task along with the baselines trained on the PointGoal task with the same amount of data (10million frames). The proposed model achieves a success rate/SPL of 0.950/0.846 as compared to 0.827/0.730 for the best baseline model on Gibson val set. We also report the performance of the proposed model trained from scratch on the PointGoal task for 10 million frames. The results indicate that the performance of ANS transferred from Exploration is comparable to ANS trained on PointGoal. This highlights a key advantage of our model. It allows us to transfer the knowledge of obstacle avoidance and control in low-level navigation across tasks, as the Local Policy and Neural SLAM module are task-invariant.

**Sample efficiency.** RL models are typically trained for more than 10 million samples. In order to compare the performance and sample-efficiency, we trained the best performing RL model (RL + Res18 + GRU + ProjDepth) for 75 million frames and it achieved a Succ/SPL of 0.678/0.486. ANS

---

[3]https://evalai.cloudcv.org/web/challenges/challenge-page/254

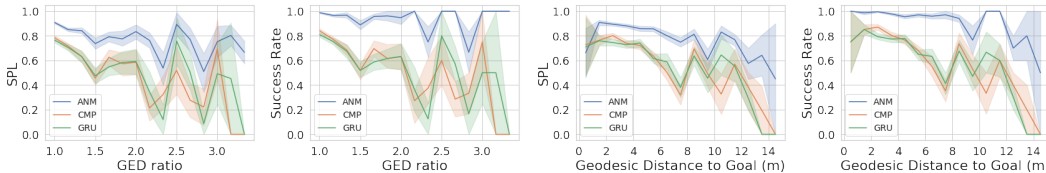

**Figure 7:** Performance of the proposed ANS model along with CMP and IL + Res18 + GRU (GRU) baselines with increase in geodesic distance to goal and increase in GED Ratio on the Gibson Val set.

Successful Trajectories                  Failure Case

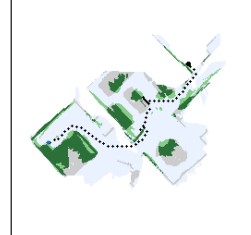 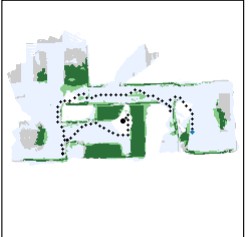 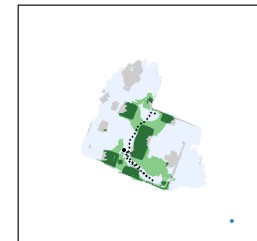

**Figure 8:** Figure showing sample trajectories of the proposed model along with the predicted map in the PointGoal task. The starting and goal locations are shown by black squares and blue circles, respectively. The ground-truth map is under-laid in grey. Map prediction is overlaid in green, with dark green denoting correct predictions and light green denoting false positives. The blue shaded region shows the explored area prediction. On the left, we show some successful trajectories which indicate that the model is effective at long distance goals with high GED ratio. On the right, we show a failure case due to mapping error.

reaches the performance of 0.789/0.703 SPL/Succ at only 1 million frames. These numbers indicate that ANS achieves $> 75\times$ speedup as compared to the best RL baseline.

**Domain and Goal Generalization:** In Table 3 (see shaded region), we evaluate all the baselines and ANS trained on the PointGoal task in the Gibson domain on the test set in Matterport domain as well as the harder goal sets in Gibson. We also transfer ANS trained on Exploration in Gibson on all the 3 sets. The results show that ANS outperforms all the baselines at all generalization sets. Interestingly, RL based methods almost fail completely on the Hard-Dist set. We also analyze the performance of the proposed model as compared to the two best baselines CMP and IL + Res18 + GRU as a function of geodesic distance to goal and GED ratio in Figure 7. The performance of the baselines drops faster as compared to ANS, especially with the increase in goal distance. This indicates that end-to-end learning methods are

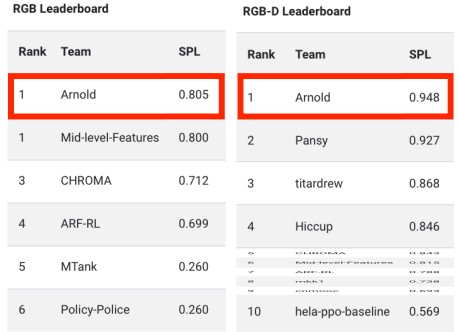

**Figure 6:** Screenshot of CVPR 2019 Habitat Challenge Results. The proposed model was submitted under code-name 'Arnold'.

effective at short-term navigation but struggle when long-term planning is required to reach a distant goal. In Figure 8, we show some example trajectories of the ANS model along with the predicted map. The successful trajectories indicate that the model exhibits strong backtracking behavior which makes it effective at distant goals requiring long-term planning. Figure 9 visualizes a trajectory in the PointGoal task show first-person observation and corresponding map predictions. Please refer to the project webpage for visualization videos.

**Habitat Challenge Results.** We submitted the ANS model to the CVPR 2019 Habitat Pointgoal Navigation Challenge. The results are shown in Figure 6. ANS was submitted under code-name 'Arnold'. ANS was the winning entry for both RGB and RGB-D tracks among over 150 submissions from 16 teams, achieving an SPL of 0.805 (RGB) and 0.948 (RGB-D) on the Test Challenge set.

# B  NOISE MODEL IMPLEMENTATION DETAILS

In order to implement the actuation and sensor noise models, we would like to collect data for navigational actions in the Habitat simulator. We use three default navigational actions: Forward:

| t=1 | t=50 | t=100 | t=150 | t=200 | t=223 |

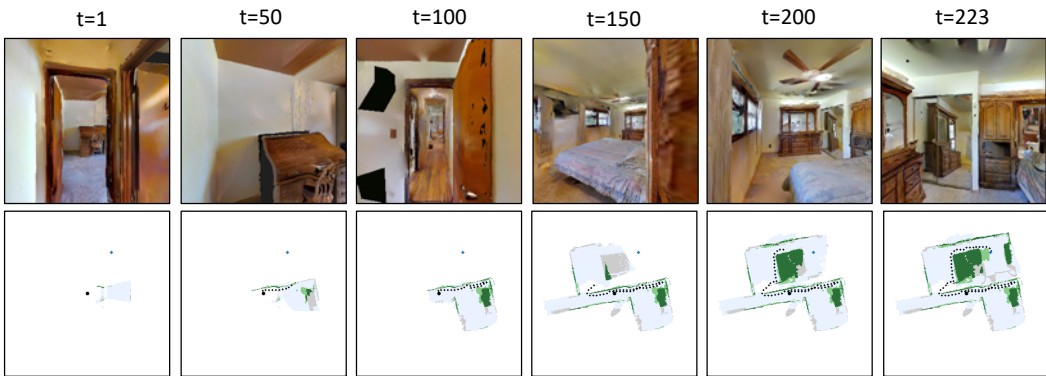

**Figure 9: Pointgoal visualization**. Figure showing sample trajectories of the proposed model along with predicted map in the Pointgoal task as the episode progresses. The starting and goal locations are shown by black squares and blue circles, respectively. Ground truth map is under-laid in grey. Map prediction is overlaid in green, with dark green denoting correct predictions and light green denoting false positives. Blue shaded region shows the explored area prediction.

move forward by 25cm, Turn Right: on the spot rotation clockwise by 10 degrees, and Turn Left: on the spot rotation counter-clockwise by 10 degrees. The control commands are implemented as $u_{Forward} = (0.25, 0, 0)$, $u_{Right} : (0, 0, -10 * \pi/180)$ and $u_{Left} : (0, 0, 10 * \pi/180)$. In practice, a robot can also rotate slightly while moving forward and translate a bit while rotating on-the-spot, creating rotational actuation noise in forward action and similarly, a translation actuation noise in on-the-spot rotation actions.

We use a Locobot [4] to collect data for building the actuation and sensor noise models. We use the pyrobot API (Murali et al., 2019) along with ROS (Quigley et al., 2009) to implement the control commands and get sensor readings. In order to get an accurate agent pose, we use an Hokuyo UST-10LX Scanning Laser Rangefinder (LiDAR) which is especially very precise in our scenario as we take static readings in 2D (Kohlbrecher et al., 2011). We install the LiDAR on the Locobot by replacing the arm with the LiDAR. We note that the Hokuyo UST-10LX Scanning Laser Rangefinder is an expensive sensor. It costs \$1600 as compared to the whole Locobot costing less than \$2000 without the arm. Using expensive sensors can improve the performance of a model, however, for a method to be scalable, it should ideally work with cheaper sensors too. In order to demonstrate the scalability of our method, we use the LiDAR only to collect the data for building noise models and not for training or deploying navigation policies in the real-world.

For the sensor estimate, we use the Kobuki base odometry available in Locobot. We approximate the LiDAR pose estimate to be the true pose of the agent as it is orders of magnitude more accurate than the base sensor. For each action, we collect 600 datapoints from both the base sensor and the LiDAR, making a total of 3600 datapoints $(600 * 3 * 2)$. We use 500 datapoints for each action to fit the actuation and sensor noise models and use the remaining 100 datapoints for validation. For each action $a$, the LiDAR pose estimates gives us samples of $p_1$ and the base sensor readings give us samples of $p'_1, i = 1, 2, \ldots, 600$. The difference between LiDAR estimates $(p_1^i)$ and control command $(\Delta u_a)$ gives us samples for the actuation noise for the action $a$: $\epsilon_{act,a}^i = p_1^i - \Delta u_a$ and difference between base sensor readings and LiDAR estimates gives us the samples for the sensor noise, $\epsilon_{sen,a}^i = p_1^{i'} - p_1^i$.

For each action $a$, we fit a separate Gaussian Mixture Model for the actuation noise and sensor noise using samples $\epsilon_{act,a}^i$ and $\epsilon_{sen,a}^i$ respectively, making a total of 6 models. We fit Gaussian mixture models with the number of components ranging from 1 to 20 for and pick the model with the highest likelihood on the validation set. Each component in these Gaussian mixture models is a multi-variate Gaussian in 3 variables, $x$, $y$ and $o$. We implement these actuation and sensor noise models in the Habitat simulator for our experiments.

---

[4]http://locobot.org

## C   NEURAL SLAM MODULE IMPLEMENTATION DETAILS

The Neural SLAM module ($f_{SLAM}$) takes in the current RGB observation, $s_t \in \mathbb{R}^{3 \times H \times W}$, the current and last sensor reading of the agent pose $x'_{t-1:t}$ and the map at the previous time step $m_{t-1} \in \mathbb{R}^{2 \times M \times M}$ and outputs an updated map, $m_t \in \mathbb{R}^{2 \times M \times M}$, and the current agent pose estimate, $\hat{x}_t$ (see Figure 2):

$$m_t, \hat{x}_t = f_{SLAM}(s_t, x'_{t-1:t}, \hat{x}_{t-1}, m_{t-1} | \theta_S, b_{t-1})$$

where $\theta_S$ denote the trainable parameters and $b_{t-1}$ denotes internal representations of the Neural SLAM module. The Neural SLAM module can be broken down into two parts, a Mapper ($f_{Map}$) and a Pose Estimator Unit ($f_{PE}$,). The Mapper outputs a egocentric top-down 2D spatial map, $p_t^{ego} \in [0,1]^{2 \times V \times V}$ (where $V$ is the vision range), predicting the obstacles and the explored area in the current observation: $p_t^{ego} = f_{Map}(s_t | \theta_M)$, where $\theta_M$ are the parameters of the Mapper. It consists of Resnet18 convolutional layers to produce an embedding of the observation. This embedding is passed through two fully-connected layers followed by 3 deconvolutional layers to get the first-person top-down 2D spatial map prediction.

Now, we would like to add the egocentric map prediction ($p_t^{ego}$) to the geocentric map from the previous time step ($m_{t-1}$). In order to transform the egocentric map to the geocentric frame, we need the pose of the agent in the geocentric frame. The sensor reading $x'_t$ is typically noisy. Thus, we have a Pose Estimator to correct the sensor reading and give an estimate of the agent's geocentric pose.

In order to estimate the pose of the agent, we first calculate the relative pose change ($dx$) from the last time step using the sensor readings at the current and last time step ($x'_{t-1}, x'_t$). Then we use a Spatial Transformation (Jaderberg et al., 2015) on the egocentric map prediction at the last frame ($p_{t-1}^{ego}$) based on the relative pose change ($dx$), $p'_{t-1} = f_{ST}(p_{t-1}^{ego} | dx)$. Note that the parameters of this Spatial Transformation are not learnt, but calculated using the pose change ($dx$). This transforms the projection at the last step to the current egocentric frame of reference. If the sensor was accurate, $p'_{t-1}$ would highly overlap with $p_t^{ego}$. The Pose Estimator Unit takes in $p'_{t-1}$ and $p_t^{ego}$ as input and predicts the relative pose change: $\hat{dx}_t = f_{PE}(p'_{t-1}, p_t^{ego} | \theta_P)$ The intuition is that by looking at the egocentric predictions of the last two frames, the pose estimator can learn to predict the small translation and/or rotation that would align them better. The predicted relative pose change is then added to the last pose estimate to get the final pose estimate $\hat{x}_t = \hat{x}_{t-1} + \hat{dx}_t$.

Finally, the egocentric spatial map prediction is transformed to the geocentric frame using the current pose prediction of the agent ($\hat{x}_t$) using another Spatial Transformation and aggregated with the previous spatial map ($m_{t-1}$) using Channel-wise Pooling operation: $m_t = m_{t-1} + f_{ST}(p_t^{ego} | \hat{x}_t)$.

Combing all the functions and transformations:

$$m_t, \hat{x}_t = f_{SLAM}(s_t, x'_{t-1:t}, m_{t-1} | \theta_S, b_{t-1})$$
$$p_t^{ego} = f_{Map}(s_t | \theta_M)$$
$$\hat{x}_t = \hat{x}_{t-1} + f_{PE}(f_{ST}(p_{t-1}^{ego} | \hat{x}_{t-1:t}), p_t^{ego} | \theta_P)$$
$$m_t = m_{t-1} + f_{ST}(p_t^{ego} | \hat{x}_t)$$
$$\text{where } \theta_M, \theta_P \in \theta_S, \quad \text{and} \quad p_{t-1}^{ego}, \hat{x}_{t-1} \in b_{t-1}$$

## D   ARCHITECTURE DETAILS

We use PyTorch (Paszke et al., 2017) for implementing and training our model. The Mapper in the Neural SLAM module consists of ResNet18 convolutional layers followed by 2 fully-connected layers trained with a dropout of 0.5, followed by 3 deconvolutional layers. The Pose Estimator consists of 3 convolutional layers followed by 3 fully connected layers. The Global Policy is a 5 layer convolutional network followed by 3 fully connected layers. We also pass the agent orientation as a separate input (not captured in the map tensor) to the Global Policy. It is processed by an Embedding layer and added as an input to the fully-connected layers. The Local Policy consists of a pretrained ResNet18 convolutional layers followed by fully connected layers and a recurrent GRU layer. In addition to the RGB observation, the Local policy receives relative distance and angle to the short-term goal as input. We bin the relative distance (bin size increasing with distance),

relative angle (5 degree bins) and current timestep (30 time step bins) before passing them through embedding layers. This kind of discretization is used previously for RL policies (Lample and Chaplot, 2017; Chaplot and Lample, 2017) and it improved the sample efficiency as compared to passing the continuous values as input directly. For a fair comparison, we use the same discretization for all the baselines as well. The short-term goal is processed using Embedding layers. For the exact architectures of all the modules, please refer to the open-source code.

## E   HYPERPARAMETER DETAILS

We train all the components with 72 parallel threads, with each thread using one of the 72 scenes in the Gibson training set. We maintain a FIFO memory of size 500000 for training the Neural SLAM module. After one step in all the environments (i.e. every 72 steps) we perform 10 updates to the Neural SLAM module with a batch size of 72. We use Adam optimizer with a learning rate of 0.0001. We use binary cross-entropy loss for obstacle map and explored area prediction and MSE Loss for pose prediction (in meters and radians). The obstacle map and explored area loss coefficients are 1 and the pose loss coefficient is 10000 (as MSE loss in meters and radians is much smaller).

The Global policy samples a new goal every 25 timesteps. We use Proximal Policy Optimization (PPO) (Schulman et al., 2017) for training the Global policy. Our PPO implementation for the Global Policy is based on Kostrikov (2018). The reward for the Global policy is the increase in coverage in $m^2$ scaled by 0.02. It is trained with 72 parallel threads and a horizon length of 40 steps (40 steps for Global policy is equivalent to 1000 low-level timesteps as Global policy samples a new goal after every 25 timesteps). We use 36 mini-batches and do 4 epochs in each PPO update. We use Adam optimizer with a learning rate of 0.000025, a discount factor of $\gamma = 0.99$, an entropy coefficient of 0.001, value loss coefficient of 0.5 for training the Global Policy.

The Local Policy is trained using binary cross-entropy loss. We use Adam optimizer with a learning rate of 0.0001 for training the Local Policy.

Input frame size is $128 \times 128$, the vision range for the SLAM module is $V = 64$, i.e. $3.2m$ (each cell is $5cm$ in length). Since there are no parameters dependent on the map size, it can be adaptive. We train with a map size of $M = 480$ (equivalent to $24m$) for training and $M = 960$ (equivalent to $48m$) for evaluation. A map of size $48m \times 48m$ is large enough for all scenes in the Gibson val set. The size of the Global Policy input is constant, $G = 240$, which means we downscale map by 2 times during training and 4 times during evaluation. All hyperparameters are available in the code.

## F   ADDITIONAL RESULTS

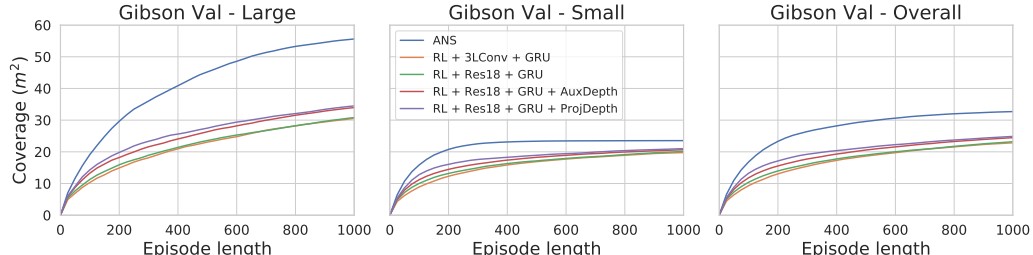

**Figure 10:** Plot showing the absolute Coverage in $m^2$ as the episode progresses for ANS and the baselines on the large and small scenes in the Gibson Val set as well as the overall Gibson Val set.

