# OpenReview forum: "Learning To Explore Using Active Neural SLAM"
_ICLR.cc/2020/Conference — Accept (Poster)_

### Official Review · AnonReviewer2 · 2019-10-13
**Official Blind Review #2**

**Rating:** 6

**Review:**

The paper describes a method for visual robot navigation in simulated environments. In terms of overall objectives and targeted reasoning, the current approaches can be roughly divided into two groups: i) learning tasks requiring high-level reasoning for navigation involving the detection and discovery of objects and their affordances and eventually also requiring to process language input, and ii) simpler navigation task involving geometry and the detection of free (navigable space): point goal, maximizing coverage etc. The former target more complex problems but the agents are more difficult (currently up to impossible) to transfer to real environments, whereas the latter directly target problems which can currently realistically used in real world scenarios.

The paper is of the second group, and addresses one of the currently investigated problems in robot navigation and mapping, namely whether learned navigation is superior to traditional planning algorithms, and whether the two different approaches can be integrated. It proposes to separate the task into long-term and short-term goals, which is not new per se, but the proposed formulation is quite interesting. In particular, the integration of the “handcrafted” planar (front propagation) into the learned framework solves a couple of issues with sample efficiency of learned methods, while still keeping some flexibility of learning over the 100% traditional approaches.

I will be upfront – I already reviewed an earlier version of this paper for NeurIPS 2019, where this paper unfortunately did not pass. I was actually a favorable reviewer at this time and was defending it. The paper has been improved since and I would be happy to see it pass. I still have a couple of questions, some of which are similar to the ones I raised in the NeurIPS review (others have been addressed since).

While I do agree that the targeted tasks might be considered less exciting then tasks involving high level semantics, I do also think that these tasks are far from solved as soon as we try to implement them in real life scenarios. I do think that the proposed paper is an interesting step forward.

The advantages of the proposed method are “bought” with a couple of key design choices, in particular the handcrafted non-differentiable long-term path planner. The downside of this is that the loss signals can’t be backpropagated through the planner, which restricts the mapping module to very simple mapping information, basically free /navigational space. End-to-end training of navigation could in principle learn to map objects and affordances which are discovered through the task and not hardcoded or even learned with supervision, which also must be known in advance. This means that the contribution is limited to simpler navigational tasks like the tested exploration and PointGoal. In contrast, other work from the literature uses differentiable planners (eg cited CMP (Gupta et al 2017), using value Iteration Networks (cited Talmar et al. 2016) which allows fine-tuning.

The mapping network, which is learned with supervision, is a general encoder-decoder network which needs to translate from projective first person views to ego-centric bird’s eye views. It thus needs to learn projective geometry from data, although projective geometry could be used as structure for the network, given camera calibration, which has been done in other work:

-	Chen et al., 2019
-	Gupta et la., 2017
-	Henriques et al., 2018
And a couple of others.

Several improvements have been made since the NeurIPS submission, some of which I had addressed in my review. The experiments are quite convincing in their comparisons with the state of the art, in particular the generalization performances:
- generalization from Gibson (training) to Matterport (testing)
- generalization from exploration (training) to PointGoal (testing).
A couple of the results have been removed from the NeurIPS submission, unfortunately, I think they should be kept in.

I appreciated the realistic sensor model fitted to real data measured with a Locobot robot, and the ablation studies, which indicated the contributions of the different planner modules and of pose estimation. The role of the short term planner has been made clearer in the new paper.

I found it interesting that the stellar performance at the Habitat AI challenge was removed from the new paper – this method (or at least a preceding version) won the challenge. But I do understand that this choice was motivated by some remarks of the NeurIPS fellow reviewers regarding the simplicity of the PointGoal task of the challenge.

A couple of less positive aspects, and questions:

On the downside, and following the remarks on literature above, I still think that the results should be compared with CMP, the main competitor of this method.
I think this is the main short coming of the paper, in particular since CMP is able to perform end-to-end training because the planner is differentiable (value iteration networks, NIPS 2016).

The literature w.r.t. to hierarchical planning is very far from exhaustive and lots of work is missing, consisting of recent work

Embodied Question Answering, Abhishek Das, Samyak Datta, Georgia Gkioxari, Stefan Lee, Devi Parikh, Dhruv Batra, CVPR 2018
(and several follow up papers)

but also quite classical work like the literature around the options framework, with the following starting point:

R.S. Sutton, D. Precup, and S. Singh. Between mdps and semi-mdps: A framework for temporal abstraction in reinforcement learning. Artificial Intelligence, 112(1):181–211, 1999.

And many other papers.

The figures 1 and 2 have been completely redone, but they are not completely clear. In particular, several intermediate representations/maps/Images are not commented or labeled, they should be annotated with the symbols from the text.

The role of the sensor output is not clear. Sensors normally provide relative positions … but the text seems to indicate absolute pose. Some details are lacking.

In “… to predict the pose change between the two maps …” it is unclear what is done here. Is this self-supervision?

The authors mention that unexplored area is considered as free space for planning. What consequences did this have in case of unexplored obstacles? I guess the problem was delegated to the local policy, which needed coping with these issues?

The last paragraph before the conclusions briefly mentions experiments and comparisons but without giving any details. This is unfortunate, since there is still space available (the paper length is 8.5 pages).


**Experience Assessment:**

I have published one or two papers in this area.

**Review Assessment: Checking Correctness Of Derivations And Theory:**

I carefully checked the derivations and theory.

**Review Assessment: Checking Correctness Of Experiments:**

I carefully checked the experiments.

**Review Assessment: Thoroughness In Paper Reading:**

I read the paper thoroughly.

---

> ### Author Response · Authors · 2019-11-14
> **Response to Reviewer #2**
>
> We thank the reviewer for valuable comments and suggestions. We address the concerns and answer the questions below:
>
>
> Regarding comparison with CMP: CMP was originally designed for the pointgoal navigation task and trained with Imitation Learning. We provided a comparison with CMP on the pointgoal task in the supplementary material. We also tried running CMP for the exploration task, however, it did not perform well in the initial set of experiments due to multiple reasons:
> a) Firstly, there is no ground-truth trajectory in the exploration task, so CMP needs to be trained with reinforcement learning rather than imitation learning. As the reviewer pointed out, CMP uses VIN as a differentiable planner and VINs do not perform as well with reinforcement learning. For example, the results in the original VIN paper show that the performance of VIN drops from 99.3% (using imitation learning) to 82.5% (using reinforcement learning) when trained on small 16x16 mazes (see Table 1 for IL results and Table 3 in Appendix for RL results: https://arxiv.org/pdf/1602.02867.pdf). We are working with orders of magnitude larger maps which makes it even more difficult for VINs to learn planning using reinforcement learning.
> b) Another complication is that both CMP and VINs were originally designed for and tested in a grid-based environment with 90-degree rotation and no motion noise. Switching to 10-degree rotations with motion noise creates aliasing effects in the map which makes it difficult to learn fine-grained navigation. We consulted with an author of CMP to ensure that our implementation is correct.
>
>
> Regarding related literature on hierarchical RL: Thank you for the suggestions. We have added the relevant literature to the related work section.
>
>
> > The figures 1 and 2 have been completely redone, but they are not completely clear.
> We have updated the figures to add more labels and correct some typos in the revised version of the manuscript.
>
>
> Regarding the role of the sensor output: We understand the source of confusion. The reviewer is correct that sensors normally provide relative positions. However, they provide the position relative to the starting position of the robot, but we need the sensor’s estimate of the position relative to the position at the last step for aligning egocentric map predictions between consecutive frames. In order to predict the pose change, we first align the egocentric map predictions at consecutive steps (using relative pose from sensor’s estimate) and then pass it through the learned pose estimator to refine the sensor’s estimate. The intuition is that by looking at the egocentric predictions of the last two frames, the pose estimator can learn to predict the small translation and/or rotation that would align them better. The pose estimator is trained using supervised learning. More details of the pose estimation model are provided in Appendix D.
>
>
> > The authors mention that unexplored area is considered as free space for planning. What consequences did this have in case of unexplored obstacles? I guess the problem was delegated to the local policy, which needed coping with these issues?
> The reviewer is correct, the local policy learns to avoid obstacles too close to the agent which are not visible in the frame and thus unexplored. We briefly discuss this in Section 6.1 Local Policy ablations.
>
>
> Regarding PointGoal navigation task: The central problem we are tackling in this paper is that of exploration in realistic settings (with realistic pose noise, etc). We moved the PointGoal results into supplementary as we felt that they distract the reader from the main message of the paper, and because we wanted to stay as close as possible to the soft 8-page limit rather than to the hard 10-page limit. We will further emphasize the PointGoal results in the main body of the paper. The supplementary material contains all relevant details already.

---

> > ### Comment · AnonReviewer2 · 2019-11-14
> > **After rebuttal**
> >
> > Thank you for your answers, they address my questions.

---

### Official Review · AnonReviewer3 · 2019-10-22
**Official Blind Review #3**

**Rating:** 3

**Review:**

This paper proposes a new architecture and policy for coverage maximization (which the authors call exploration).  Overall the paper is well written, but I have some major concerns. However I am not an expert in navigation / robotics so i have given myself the lowest confidence for this paper.

My highest level concern is that this approach seems extremely complicated (eg Figs 1 and 2), as well as employing several sub-algorithms as part of the procedure (eg Fast Marching Method). It's not clear to me why any of the components are necessary, though I do appreciate the ablation study. But even within that ablation not all components are ablated (e.g., why GRU units?).  My experience suggests that extremely complicated architectures such as this one are brittle and don't generalize (and it goes against Sutton's 'bitter lesson'). The fact that the experiments are so small does not help. Perhaps more challenging domains would yield negative results. Further, how tuned are the baselines? And it seems that the baselines are general RL agents and not optimized for coverage maximization like this architecture. The authors say " We will also open-source the code", has this been done? Open-sourcing would help others reproduce the results since as it stands I think this is too complicated to be reproduced. The level of intricacy makes me think that perhaps this paper is more suited to a robotics conference.

Secondly, the paper mentions exploration a lot, but it's not clear to me how this is a principled exploration strategy. Exploration is not in fact defined as "visit as much area as possible" or "maximize the coverage in a fixed time budget", as the authors suggest. In fact the sentences "We follow the exploration task setup proposed by Chen et al. 2019 where the objective is to maximize the coverage in a fixed time budget. [The] coverage is defined as the total area in the map known to be traversable" appears twice in this manuscript. Exploration is better defined within the context of the explore-exploit tradeoff, whereby an agent must sometimes take sub-optimal actions in order to learn more about the environment in the hope of possibly increasing it's long-term return. Conflating 'coverage-maximization' and exploration is confusing. I think the paper should be rewritten to de-emphasize exploration and instead talk about coverage-maximization, which is more accurate.

"Exploration has also been studied more generally in RL for faster training (Schmidhuber, 1991)." I certainly would *not* cite Schmidhuber 91 as the canonical reference of exploration in RL. Far, far, more appropriate would be either the Sutton+Barto RL book (which doesn't do a great job covering exploration but is at least a decent overall reference) or the works of Auer 2002 and Jaksch et al 2010, and related papers. The Schmidhuber citation should be removed and replaced with a few that actually make sense in this context.

I don't understand how the goals (especially long-term) are generated and trained. Is the long-term goal trained using the reward signal? This is not properly explained.

"and summarize major these below" typo, probably should be themes or theses?

"agnet pose" typo.

**Experience Assessment:**

I do not know much about this area.

**Review Assessment: Checking Correctness Of Derivations And Theory:**

N/A

**Review Assessment: Checking Correctness Of Experiments:**

I assessed the sensibility of the experiments.

**Review Assessment: Thoroughness In Paper Reading:**

I read the paper at least twice and used my best judgement in assessing the paper.

---

> ### Author Response · Authors · 2019-11-14
> **Response to Reviewer #3**
>
> We thank the reviewer for helpful feedback. We address your concerns below.
>
> Q: Regarding the complexity of our method.
> A: We argue that our method is not very complicated and easy to train due to its modularity. Our experiments indicate that the baseline methods based on end-to-end learning do not perform well. In order to tackle the limitations of end-to-end learning, we break down the problem and propose a modular hierarchical model. Each module is simple and easy to train as they do not need to be trained jointly. The importance of each module is shown in the ablation experiments.
>
> Regarding Fast Marching Method: It is a simple shortest path planning algorithm which we implement using a few lines of Python code using an off-the-shelf package. One can also use other shortest path algorithms, such as A* or Djikstra’s instead of the Fast Marching Method.
>
> Regarding GRUs: Recurrent layers are commonly used in navigation models. The motivation is that the agent needs to have some memory of prior observations to navigate effectively. In our case, we need memory to get feedback of obstacles not visible in the current frame. Among the types of recurrent units, we found that both LSTMs and GRUs gave similar performance, and we chose GRU as it was slightly faster.
>
> Q: Experiments are small, more challenging domains would yield negative results.
> A: On the contrary, we believe our experimental setup is as realistic and challenging as it gets, significantly more so than any prior work in the area:
> Simulation environments are scans of real environments, so they retain the visual complexity.
> Actuation noise models are derived from real robot runs (vs no noise, or artificial Gaussian noise in past works).
> The model generalizes to new Matterport domains out-of-the-box (Tables 1 and 3).
> The model also works in the real-world (which in our opinion is the most challenging and useful domain).
>
> Regarding the definition of exploration: ‘Exploration’ is a slightly overloaded term having different meanings in the context of navigation and in the context of exploration-exploitation trade-off in RL. We would like to point out that the definition of exploration used in the paper is not our definition but has been used in the navigation literature for over two decades [for eg. 1 - 6]. The same definition of exploration is also used in recent machine learning papers tackling exploration in the context of navigation, for example, Chen et al. [7] (ICLR 2019), Fang et al. [8] (CVPR 2019). We use the same definition to keep the terminology consistent in the literature.
>
> Regarding references on exploration in RL, thanks for pointing out the error. We agree that the suggested references are much more relevant, and have revised the paper to correct it. The Schmidhuber, 91 reference was originally added for curiosity-based exploration in RL but was incorrectly referenced over revisions.
>
> Thanks for pointing out the typos, those have been corrected in the revision.
>
> [1] B. Yamauchi, “A Frontier Based Approach for Autonomous Exploration,” in Proceedings of the IEEE International Symposium on Computational Intelligence in Robotics and Automation (CIRA), 1997, pp. 146–151.
>
> [2] F. Amigoni and A. Gallo, “A Multi-Objective Exploration Strategy for Mobile Robots,” in Proceedings of the IEEE International Conference on Robotics and Automation (ICRA), 2005, pp. 3861–3866
>
> [3] W. Burgard, M. Moors, C. Stachniss, and F. Schneider, “Coordinated Multi-Robot Exploration,” IEEE Transactions on Robotics, vol. 21, no. 3, pp. 376– 386, 2005.
>
> [4] R. Sim and N. Roy, “Global A-Optimal Robot Exploration in SLAM,” in Proceedings of the IEEE International Conference on Robotics and Automation (ICRA), April 2005, pp. 661–666.
>
> [5] F. Amigoni, “Experimental Evaluation of Some Exploration Strategies for Mobile Robots,” in Proceedings of the IEEE International Conference on Robotics and Automation (ICRA), 2008, pp. 2818– 2823.
>
> [6] Dirk Holz, Nicola Basilico, Francesco Amigoni, and Sven Behnke. Evaluating the efficiency of frontier-based exploration strategies. In ISR 2010 (41st International Symposium on Robotics) and ROBOTIK 2010 (6th German Conference on Robotics), pages 1–8. VDE, 2010.
>
> [7] Tao Chen, Saurabh Gupta, and Abhinav Gupta. Learning exploration policies for navigation. In ICLR, 2019
>
> [8] Kuan Fang, Alexander Toshev, Li Fei-Fei, and Silvio Savarese. Scene memory transformer for embodied agents in long-horizon tasks. In CVPR, 2019.

---

### Official Review · AnonReviewer1 · 2019-10-29
**Official Blind Review #1**

**Rating:** 8

**Review:**

The paper describes ANM, active neural mapping, to learn policies for efficiently exploring 3d environments. The paper combines classical methods with learning based approaches, allowing the final system to work competitively with raw sensory inputs without requiring unreasonable amounts of training samples.

I think this is a well-written "ML-systems paper" and I'm especially happy that real-world aspects of mobile robots are taken into account. I was able to follow the overall idea of the approach as well as the description of the three components. I also think that the experiments are well done, showing convincingly ANMs competitive performance and demonstrate, through the ablation studies, the importance of its constituting parts.

**Experience Assessment:**

I have read many papers in this area.

**Review Assessment: Checking Correctness Of Derivations And Theory:**

I assessed the sensibility of the derivations and theory.

**Review Assessment: Checking Correctness Of Experiments:**

I assessed the sensibility of the experiments.

**Review Assessment: Thoroughness In Paper Reading:**

I read the paper at least twice and used my best judgement in assessing the paper.

---

> ### Author Response · Authors · 2019-11-14
> **Thank you.**
>
> We thank the reviewer for the motivational feedback! We are glad that our efforts to tackle the challenges involved in the real-world aspects of mobile robotics and navigation are appreciated.

---

### Author Response · Authors · 2019-11-14
**Author Response**

We thank the reviewers for the helpful feedback. The reviewers have appreciated our realistic experimental design, strong generalization results, and ablation studies. They found our experiments convincing in their comparisons with the state of the art. We are glad that our effort to tackle real-world aspects of the exploration problem in the context of navigation was appreciated.

The reviewers had requested some clarifications and had some suggestions about related work. We provide these clarifications in individual responses to the reviewers below and have made minor revisions in the paper accordingly.

---

### Decision · Program_Chairs · 2019-12-19

**Decision:**

Accept (Poster)

**Comment:**

The paper presents a method for visual robot navigation in simulated environments. The proposed method combines several modules, such as mapper, global policy, planner, local policy for point-goal navigation. The overall approach is reasonable and the pipeline can be modularly trained. The experimental results on navigation tasks show strong performance, especially in generalization settings.